# Localization and Tracking of Discrete Mobile Scatterers in Vehicular Environments Using Delay Estimates [note 1]

**DOI:** 10.3390/s19214802

**Published:** 2019-11-05

**Authors:** Martin Schmidhammer, Christian Gentner, Benjamin Siebler, Stephan Sand

**Affiliations:** German Aerospace Center (DLR), Institute of Communications and Navigation, 82234 Wessling, Germany; christian.gentner@dlr.de (C.G.); benjamin.siebler@dlr.de (B.S.); stephan.sand@dlr.de (S.S.)

**Keywords:** mulitlateration, localization, nonlinear least-squares, Levenberg–Marquardt, tracking, extended Kalman filter, Bayesian performance bounds, posterior Cramér–Rao lower bound

## Abstract

This paper describes an approach to detect, localize, and track moving, non-cooperative objects by exploiting multipath propagation. In a network of spatially distributed transmitting and receiving nodes, moving objects appear as discrete mobile scatterers. Therefore, the localization of mobile scatterers is formulated as a nonlinear optimization problem. An iterative nonlinear least squares algorithm following Levenberg and Marquardt is used for solving the optimization problem initially, and an extended Kalman filter is used for estimating the scatterer location recursively over time. The corresponding performance bounds are derived for both the snapshot based position estimation and the nonlinear sequential Bayesian estimation with the classic and the posterior Cramér–Rao lower bound. Thereby, a comparison of simulation results to the posterior Cramér–Rao lower bound confirms the applicability of the extended Kalman filter. The proposed approach is applied to estimate the position of a walking pedestrian sequentially based on wideband measurement data in an outdoor scenario. The evaluation shows that the pedestrian can be localized throughout the scenario with an accuracy of 0.8 m at 90% confidence.

## 1. Introduction

With the trends of increasing urbanization and increasing automation in road transportation, the demand for improvements in vehicular safety technologies is steadily growing. In particular the mixed-traffic environment shared by many different users including cars, motorcycles, cyclists, and pedestrians is challenging for any means of automated transport. To safely route through vehicular environments, therefore, timely and reliable information about other road users is required. In this regard, the exchange of user specific information, like position and velocity, enhances safety on roads by supporting mutual awareness [1,2,3]. This cooperative approach requires road users to be equipped with actively probing devices to determine their user specific information and to exchange the data. However, many road users do not carry any devices, i.e., they are non-cooperative. Thus, for a reliable and comprehensive situational awareness, further methods and sensor technologies are needed, accounting for non-cooperative road users. With regard to current automated and autonomous vehicles, the perception of their surrounding environment mainly relies on locally mounted sensors, including radar and lidar sensors, as well as camera based systems [4]. Due to physical properties, however, locally mounted perception sensors exhibit a series of critical limitations, like the limited performance in adverse lighting conditions [3]. Thus, to improve the reliability and to extend the awareness range of the local perception sensors, infrastructure based systems have been suggested, for instance based on dedicated radar sensors [5] and cameras [6]. Apart from remaining technical challenges, the main drawback of these systems is their deployment. For sufficiently sensing only a limited area like an urban intersection, several sensors would have to be mounted, which results in high deployment and maintenance costs.

Therefore, a localization system is introduced in [7], which reuses signals from vehicular communications infrastructure for passive radar application. It is assumed that road users and other objects affect the radio spectrum by inducing delayed and Doppler shifted MPC. This means that the characteristics of the MPC correspond to location and dynamics of the road users. Thus, sensing the wireless propagation characteristics between the links of existing communication networks allows for detecting and localizing road users. Similarly, the authors in [8] propose the usage of existing vehicular radio links for detecting and localizing road users. In addition to static links from communication infrastructure, they propose to include also mobile devices as possible network nodes. The localization accuracy of both systems strongly depends on precise location information of the individual network nodes. Thus, an incorporation of mobile nodes requires to account for location uncertainties, which can degrade the localization performance.

The proposed sensing systems of [7] and [8] follow the idea of PCL, i.e., the usage of arbitrary signals as illuminators of opportunity [9]. With its origin in aeronautics, PCL was mainly used for observing targets far from the sensing network [9,10,11,12]. The localization requirements, for instance, from aeronautic and maritime applications allow high integration times and low bandwidths. Recently, the interest in passive localization approaches for short-range and passive indoor localization is increasing [13,14,15]. The main challenge of short-range PCL are the dynamics of targets moving in the proximity of the sensing network. Thereby, signals from Wi-Fi access points with comparatively large bandwidths are beneficial for localization, since lower integration times are required [7,13]. In this regard, the authors of [16] propose to use even ultra-wideband signals for PCL. Furthermore, they introduce different target tracking algorithms and analyze the algorithms in simulations. Another localization approach was introduced in [17], applying an iterative, nonlinear least-squares approach for position estimation. The applicability of the approach is demonstrated using wideband measurement data.

On the basis of [17], this work extends the localization approach by sequential target tracking, namely an EKF. For the evaluation of the tracking algorithm, the PCRLB is derived. Finally, the proposed localization approach is applied to wideband measurement data. The main contributions of this paper are:The signal processing for a localization approach to localize moving, non-cooperative objects recursively, using delay estimates from a network of spatially distributed transmitting and receiving nodes.The derivation of performance bounds including the CRLB on position estimation and the PCRLB on nonlinear sequential Bayesian estimation.The validation of the applicability of an EKF for the introduced localization problem via Monte Carlo simulations and a comparison to the PCRLB.The application of the proposed localization approach to wideband measurement data of an outdoor experiment for localizing a walking pedestrian.

The remainder of this paper is structured as follows. Section 2 introduces the network structure together with the measurement model used for position estimation. In Section 3, the signal processing is described in order to localize mobile scatterers, which comprises a procedure to extract the measurement vector, a snapshot based position estimator, as well as a nonlinear sequential Bayesian estimator. Corresponding performance bounds for positioning and for nonlinear sequential Bayesian estimation are derived in Section 4. Based on an exemplary measurement setup, the localization approach is evaluated in Section 5, first theoretically by using the derived performance bounds and second by applying the approach to channel measurements. Finally, Section 6 concludes the paper by summarizing the findings.

## 2. Network and Measurement Model

This section introduces the network structure and the measurement model used for the position estimation. Refer to a widely distributed network of *K* transmitting and *L* receiving nodes, where both transmitting and receiving nodes are assumed to be static at known locations $r_{k}^{\mathrm{Tx}}$, k∈{1,⋯,K}, and $r_{l}^{\mathrm{Rx}}$, l∈{1,⋯,L}, respectively. Receiving nodes can be collocated with transmitting nodes or individually placed. The network link configuration determines the index set P, where each link (k,l)∈P is composed of the *k*-th transmitting node and the *l*-th receiving node. Accordingly, fully meshed networks result in a maximum number of |P|=KL links.

Each transmitting node emits known signals $s_{k}\left(t\right)$ with period $T_{p}$, which allows for measuring the CIR at the receivers [18]. Thereby, the received signal of each link in the network is modeled as a superposition of a finite number of scaled and delayed replica of the transmit signal. These comprise the LoS and a discrete number of MPCs due to reflection and scattering. Following [19], the MPCs are further differentiated according to the dynamics of the scattering objects, i.e., static and mobile. Eventually, the CIR for a pair of transmitting and receiving nodes ${\mathrm{Tx}}_{k}$ and ${\mathrm{Rx}}_{l}$ can be expressed as
(1)$$h_{kl}\left(t,\tau \right)=h_{kl}^{\mathrm{LoS}}\left(t,\tau \right)+h_{kl}^{S}\left(t,\tau \right)+h_{kl}^{M}\left(t,\tau \right),$$ where $h_{kl}^{\mathrm{LoS}}\left(t,\tau \right)$, $h_{kl}^{S}\left(t,\tau \right)$, and $h_{kl}^{M}\left(t,\tau \right)$ denote the contribution of the LoS, the sum of $R_{kl}$ discrete static MPC and of $Q_{kl}$ discrete mobile MPC to the CIR, respectively.

The static network allows for attributing mobile MPC to moving scattering objects in the observed area. Here, the number of mobile MPC is assumed to be identical for each link in the network, i.e., $Q_{kl}=Q$, and the mobile MPC can be uniquely assigned to the moving objects. Consequently, for time and phase synchronized transmitting and receiving nodes, discrete mobile scatterers are contributing to the CIR by
(2)$$h_{kl}^{M}\left(t,\tau \right)=\sum_{q=1}^{Q}\alpha_{klq}^{M}\left(t\right)e^{-j2\pi f_{c}\tau_{kl}^{M}\left(r_{q}\left(t\right)\right)}\delta \left(\tau -\tau_{kl}^{M}\left(r_{q}\left(t\right)\right)\right),$$ with $\alpha_{klq}^{M}\left(t\right)$ and $\tau_{kl}^{M}\left(r_{q}\left(t\right)\right)$ representing the complex amplitude and the propagation delay associated with the *q*th mobile scatterer at location $r_{q}\left(t\right)={\left[x_{q}\left(t\right),y_{q}\left(t\right)\right]}^{T}$, and δ(·) representing the Dirac function. The corresponding delay-induced phase shift for center frequency $f_{c}$ is expressed by the term $e^{-j2\pi f_{c}\tau_{klq}^{M}\left(r_{q}\left(t\right)\right)}$. For convenience, the notation for time dependence will be omitted in the remainder of this paper and only applied where explicitly needed.

For any pair of transmitting and receiving node, the propagation delay induced by scatterer *q* is determined by the physical propagation path from transmitter to scatterer and from scatterer to receiver. Thus, given the distances $d_{kq}^{\mathrm{Tx}}$ and $d_{lq}^{\mathrm{Rx}}$ between scatterer and ${\mathrm{Tx}}_{k}$ and ${\mathrm{Rx}}_{l}$, respectively, the propagation delay can be expressed as
(3)$$\begin{array}{cc} \tau_{kl}^{M}\left(r_{q}\right) & =\frac{1}{c}\left(d_{kq}^{\mathrm{Tx}}+d_{lq}^{\mathrm{Rx}}\right)\\ \; & =\frac{1}{c}\left(∥r_{q}-r_{k}^{\mathrm{Tx}}∥+∥r_{q}-r_{l}^{\mathrm{Rx}}∥\right), \end{array}$$ where *c* denotes the speed of light and the operator ∥·∥ denotes the Euclidean norm. For |P| network links, linear independent propagation delays induced by the *q*th scatterer compose a vector $\tau_{q}$, with each vector element $\tau_{klq}^{M}=\tau_{kl}^{M}\left(r_{q}\right)$. Thus, for the mapping $h\left(r_{q}\right)=\tau_{q}$ with each element defined according to Equation (Equation 3), the measurement model is given by
(4)$${\hat{\tau }}_{q}=\tau_{q}+w_{q}=h\left(r_{q}\right)+w_{q},$$ with ${\hat{\tau }}_{q}$ as corresponding vector of measured delays and $w_{q}$ as zero-mean white Gaussian noise with covariance matrix $R_{q}$ defined as diagonal matrix with elements ${\left\{\sigma_{klq}^{2}\right\}}_{(k,l)\in P}$.

## 3. Localization and Tracking

As shown above, propagation delays of time-variant MPC inherently contain location information of mobile scatterers being reflected in the CIR. Besides time-variant MPC, the CIR of each link in the network also comprise a static LoS and static MPC as stated in Equation (Equation 1). Thus, for localizing and tracking mobile scatterers, time-variant channel components need to be initially extracted. Therefore, the proposed localization approach is composed of three stages—first, a calibration stage to identify and characterize the static LoS and MPC, second, an estimation stage to extract mobile MPC, and third, a tracking stage to estimate the scatterer position recursively.

Both calibration and estimation stage rely on estimates of channel parameters, including complex amplitude and propagation delays of the LoS and MPC. To estimate and track the channel parameters of the CIR, KEST is used [18].

### 3.1. Calibration Stage

In order to deduce location information of moving objects from CIR measurements, the propagation effects of the static environment need to be known. Therefore, the channel of every link in the network is initially observed over a calibration period $T_{\mathrm{cal}}$. During this period, the environment is assumed to be devoid of any moving object, i.e., Q=0. With $T_{g}$ as the time interval between two adjacent CIR measurements, a total of $\left\lfloor T_{\mathrm{cal}}/T_{g}\right\rfloor$ consecutive CIR are collected for calibration. Given the set of recorded CIR, first, the channel parameters are estimated using KEST. Second, the parameter estimates are clustered with regard to amplitude and delay [20]. For static environments, the resulting clusters correspond to the LoS and to static MPC. Thereby, the vectors ${\bar{\tau }}_{klr}$ and ${\bar{\sigma }}_{klr}$, $r\in \{0,\cdots ,R_{kl}\}$, contain cluster mean and standard deviation for the LoS (r=0) and $R_{kl}$ static MPC. Eventually, the vectors ${\bar{\tau }}_{kl}={\left[{\bar{\tau }}_{kl0},\cdots ,{\bar{\tau }}_{klR_{kl}}\right]}^{T}$ and ${\bar{\sigma }}_{kl}={\left[{\bar{\sigma }}_{kl0},\cdots ,{\bar{\sigma }}_{klR_{kl}}\right]}^{T}$ uniquely characterize the static propagation environment between *k*-th transmitter ${\mathrm{Tx}}_{k}$ and *l*-th receiver ${\mathrm{Rx}}_{l}$. Note that, even though the static environment is only typically changing very slowly, any modification in the propagation condition requires to recalibrate the system. Particularly, objects with strongly reflecting characteristics, will change the conditions significantly, as for example, the placement of a car in the network environment.

### 3.2. Estimation Stage

Other than during the calibration period, now, additional objects can move within the observed environment. Thus, the CIR and therefore also the estimated channel parameter comprise the propagation effects of moving objects. Similar to before, KEST is applied for estimating channel parameters from incoming CIR. The resulting vector of propagation delay estimates is composed as
(5)$${\hat{\tau }}_{kl}^{\mathrm{full}}={\left[\underset{\mathrm{LoS}\;\mathrm{and}\;\mathrm{static}\;\mathrm{MPC}}{\underset{︸}{{\hat{\tau }}_{kl0}^{S},\cdots ,{\hat{\tau }}_{klR_{kl}}^{S}}},{\hat{\tau }}_{kl1}^{M},\cdots ,{\hat{\tau }}_{klQ}^{M}\right]}^{T}.$$

Consequently, extracting mobile MPCs from ${\hat{\tau }}_{kl}^{\mathrm{full}}$ means to sort out static components. The sorting is based on ${\bar{\tau }}_{kl}$ determined in the preceding calibration stage. Particularly, all elements of ${\hat{\tau }}_{kl}^{\mathrm{full}}$ lying in an interval of ${\bar{\tau }}_{klp}\pm 3{\bar{\sigma }}_{klp}$ are excluded. Here, the 3-σ interval ensures that delay estimates assigned as static are not considered as mobile MPC with a probability higher than 99%, assuming that the previously determined amount of static MPCs remains constant. Eventually, the measurement vector of mobile MPCs for the link ${\mathrm{Tx}}_{k}$ and ${\mathrm{Rx}}_{l}$ is
(6)$${\hat{\tau }}_{kl}={\left[{\hat{\tau }}_{kl1}^{M},\cdots ,{\hat{\tau }}_{klQ}^{M}\right]}^{T}.$$

Associated elements of ${\hat{\tau }}_{kl}$ to scatterer *q* are rearranged over all links in the network, P, which finally determine the measurement vector ${\hat{\tau }}_{q}$.

Based on the measurement model in Equation (Equation 4), scatterer *q* can be localized applying maximum likelihood estimation. In particular, a weighted nonlinear least-squares approach [21,22] is used to minimize the cost function
(7)$$L\left(r_{q}\right)={\left({\hat{\tau }}_{q}-\tau_{q}\right)}^{T}R_{q}^{-1}\left({\hat{\tau }}_{q}-\tau_{q}\right),$$ with respect to the unknown position $r_{q}$, which is expressed as
(8)$${\hat{r}}_{q}=\underset{r_{q}}{\arg\;\min}\;L\left(r_{q}\right).$$

The two-dimensional, nonlinear optimization problem of Equation (Equation 8) needs to be solved by an iterative approach, since no closed-form solution is existing. Thus, in order to minimize the cost function in Equation (Equation 7), the proposed localization approach applies the Levenberg–Marquardt algorithm [22] due to high robustness and fast convergence characteristics. Particularly, the iterative procedure can be written as
(9)$$\begin{array}{c} r_{q}^{\left(i+1\right)}=r_{q}^{\left(i\right)}+{\left(J^{T}\left(r_{q}^{\left(i\right)}\right)R_{q}^{-1}J\left(r_{q}^{\left(i\right)}\right)+\lambda^{\left(i\right)}I\right)}^{-1}\\ J^{T}\left(r_{q}^{\left(i\right)}\right)R_{q}^{-1}\left({\hat{\tau }}_{q}-\tau_{q}^{\left(i\right)}\right), \end{array}$$ with $J(r_{q})$, I and $\lambda^{\left(i\right)}$ denoting the Jacobian matrix, the identity matrix, and the dampening parameter for iteration step *i*, respectively. Individual elements of vector $\tau_{q}^{\left(i\right)}$ are calculated according to Equation (Equation 3) as $\tau_{kl}^{M}\left(r_{q}^{\left(i\right)}\right)$. The elements of Jacobian matrix $J\left(r_{q}\right)\in R^{|P|\times 2}$, i.e., the partial derivatives of $\tau_{q}$ with respect to $r_{q}$, are given as
(10)$$J\left(r_{q}\right)=\left[\begin{array}{cc} \frac{x_{q}-x_{1}^{\mathrm{Tx}}}{d_{1q}^{\mathrm{Tx}}}+\frac{x_{q}-x_{1}^{\mathrm{Rx}}}{d_{1q}^{\mathrm{Rx}}} & \frac{y_{q}-y_{1}^{\mathrm{Tx}}}{d_{1q}^{\mathrm{Tx}}}+\frac{y_{q}-y_{1}^{\mathrm{Rx}}}{d_{1q}^{\mathrm{Rx}}}\\ \vdots & \vdots \\ \frac{x_{q}-x_{K}^{\mathrm{Tx}}}{d_{Kq}^{\mathrm{Tx}}}+\frac{x_{q}-x_{1}^{\mathrm{Rx}}}{d_{1q}^{\mathrm{Rx}}} & \frac{y_{q}-y_{K}^{\mathrm{Tx}}}{d_{Kq}^{\mathrm{Tx}}}+\frac{y_{q}-y_{1}^{\mathrm{Rx}}}{d_{1q}^{\mathrm{Rx}}}\\ \frac{x_{q}-x_{1}^{\mathrm{Tx}}}{d_{1q}^{\mathrm{Tx}}}+\frac{x_{q}-x_{2}^{\mathrm{Rx}}}{d_{2q}^{\mathrm{Rx}}} & \frac{y_{q}-y_{1}^{\mathrm{Tx}}}{d_{1q}^{\mathrm{Tx}}}+\frac{y_{q}-y_{2}^{\mathrm{Rx}}}{d_{2q}^{\mathrm{Rx}}}\\ \vdots & \vdots \\ \frac{x_{q}-x_{K}^{\mathrm{Tx}}}{d_{Kq}^{\mathrm{Tx}}}+\frac{x_{q}-x_{2}^{\mathrm{Rx}}}{d_{2q}^{\mathrm{Rx}}} & \frac{y_{q}-y_{K}^{\mathrm{Tx}}}{d_{Kq}^{\mathrm{Tx}}}+\frac{y_{q}-y_{2}^{\mathrm{Rx}}}{d_{2q}^{\mathrm{Rx}}}\\ \vdots & \vdots \\ \frac{x_{q}-x_{1}^{\mathrm{Tx}}}{d_{1q}^{\mathrm{Tx}}}+\frac{x_{q}-x_{L}^{\mathrm{Rx}}}{d_{Lq}^{\mathrm{Rx}}} & \frac{y_{q}-y_{1}^{\mathrm{Tx}}}{d_{1q}^{\mathrm{Tx}}}+\frac{y_{q}-y_{L}^{\mathrm{Rx}}}{d_{Lq}^{\mathrm{Rx}}}\\ \vdots & \vdots \\ \frac{x_{q}-x_{K}^{\mathrm{Tx}}}{d_{Kq}^{\mathrm{Tx}}}+\frac{x_{q}-x_{L}^{\mathrm{Rx}}}{d_{Lq}^{\mathrm{Rx}}} & \frac{y_{q}-y_{K}^{\mathrm{Tx}}}{d_{Kq}^{\mathrm{Tx}}}+\frac{y_{q}-y_{L}^{\mathrm{Rx}}}{d_{Lq}^{\mathrm{Rx}}} \end{array}\right].$$

### 3.3. Tracking Stage

The previous stage provides an approach to localize a moving scatterer based on delay measurements at one specific time instance *t*. Since the goal of this work is to localize moving scatterers, it is reasonable to additionally take into account the mobility of the object by filtering the state evolution over time. For any mobile scatterer *q*, the state is defined by
(11)$$x_{q}\left(t_{n}\right)={\left[r_{q}^{T}\left(t_{n}\right),v_{q}^{T}\left(t_{n}\right)\right]}^{T},$$ where $r_{q}\left(t_{n}\right)$ and $v_{q}\left(t_{n}\right)={\left[v_{q,x}\left(t_{n}\right),v_{q,y}\left(t_{n}\right)\right]}^{T}$ denote the position and the velocity of the scatterer at time instant $t_{n}$. To describe the state evolution from time instant $t_{n-1}$ to time instant $t_{n}$, a transition model is applied. Accounting for the mobility of scatterers induced by different road users, such as cars, bikes, and pedestrians, a white noise acceleration model is used [23]. The state equation results in
(12)$$x_{q}\left(t_{n}\right)=A_{q}x_{q}\left(t_{n-1}\right)+n_{q}\left(t_{n}\right),$$ with transition matrix $A_{q}$ and zero-mean white Gaussian process noise $n_{q}\left(t_{n}\right)$ with covariance matrix $Q_{q}$. Being $T_{g}=t_{n}-t_{n-1}$, the transition matrix is given by
(13)$$A_{q}=\left[\begin{array}{cccc} 1 & 0 & T_{g} & 0\\ 0 & 1 & 0 & T_{g}\\ 0 & 0 & 1 & 0\\ 0 & 0 & 0 & 1 \end{array}\right]$$ and the covariance matrix by
(14)$$Q_{q}=\left[\begin{array}{cccc} \sigma_{q}^{2}\frac{T_{g}^{3}}{3} & 0 & \sigma_{q}^{2}\frac{T_{g}^{2}}{2} & 0\\ 0 & \sigma_{q}^{2}\frac{T_{g}^{3}}{3} & 0 & \sigma_{q}^{2}\frac{T_{g}^{2}}{2}\\ \sigma_{q}^{2}\frac{T_{g}^{2}}{2} & 0 & \sigma_{q}^{2}T_{g} & 0\\ 0 & \sigma_{q}^{2}\frac{T_{g}^{2}}{2} & 0 & \sigma_{q}^{2}T_{g} \end{array}\right],$$ with $\sigma_{q}^{2}$ as process noise intensity of physical dimension $\left[m^{2}/s3\right]$, which needs to be set according to application requirements [23].

Given state vector $x_{q}\left(t_{n}\right)$, the nonlinear mapping of the measurement model in Equation (Equation 4) can be expressed as $h\left(r_{q}\left(t_{n}\right)\right)=h\left(x_{q}\left(t_{n}\right)\right)$, equivalently. Accordingly, the measurement model is given by
(15)$$z_{q}\left(t_{n}\right)=h\left(x_{q}\left(t_{n}\right)\right)+w_{q}\left(t_{n}\right).$$

Due to the nonlinearity of the measurement model, a recursive Bayesian filter is required, which is able to handle general nonlinear problems. A common implementation of such recursive Bayesian filters is the EKF. Given the nonlinear measurement model in Equation (Equation 4) and the linear state model in Equation (Equation 12), the EKF results in the following set of equations: (16)$$\begin{array}{cc} {\hat{x}}_{q}\left(t_{n}|t_{n-1}\right) & =A_{q}{\hat{x}}_{q}\left(t_{n-1}|t_{n-1}\right),\\ P_{q}\left(t_{n}|t_{n-1}\right) & =A_{q}P_{q}\left(t_{n-1}|t_{n-1}\right)A_{q}^{T}+Q_{q},\\ K(t_{n}) & =P_{q}\left(t_{n}|t_{n-1}\right)H_{q}^{T}\left(t_{n}\right){\left(H_{q}\left(t_{n}\right)P_{q}\left(t_{n}|t_{n-1}\right)H_{q}^{T}\left(t_{n}\right)+R_{q}\left(t_{n}\right)\right)}^{-1},\\ {\hat{x}}_{q}\left(t_{n}|t_{n}\right) & ={\hat{x}}_{q}\left(t_{n}|t_{n-1}\right)+K\left(t_{n}\right)\left({\hat{\tau }}_{q}\left(t_{n}\right)-h\left({\hat{x}}_{q}\left(t_{n}|t_{n-1}\right)\right)\right),\\ P_{q}\left(t_{n}|t_{n}\right) & =P_{q}\left(t_{n}|t_{n-1}\right)-K\left(t_{n}\right)H_{q}\left(t_{n}\right)P_{q}\left(t_{n}|t_{n-1}\right), \end{array}$$ where $K(t_{n})$ is the Kalman gain, ${\hat{x}}_{q}\left(t_{n}|t_{n-1}\right)$ and $P_{q}\left(t_{n}|t_{n-1}\right)$ are the predicted state and covariance matrix at time $t_{n}$, and ${\hat{x}}_{q}\left(t_{n}|t_{n}\right)$ and $P_{q}\left(t_{n}|t_{n}\right)$ are the corrected state estimate and covariance matrix after the measurement update. The local linearization of the measurement model around ${\hat{x}}_{q}\left(t_{n}|t_{n-1}\right)$ is denoted in the observation matrix $H_{q}\left(t_{n}\right)$. Since the elements of measurement vector $\tau_{q}\left(t_{n}\right)$ only depend on position and not on velocity, the linearized observation matrix consists of the Jacobian in Equation (Equation 10) at the predicted position ${\hat{r}}_{q}\left(t_{n}|t_{n-1}\right)$ and of a |P|×2 zero matrix 0, written as
(17)$$H_{q}\left(t_{n}\right)=\left[J({\hat{r}}_{q}\left(t_{n}|t_{n-1}\right)),0\right].$$

## 4. Performance Bounds

For evaluating estimators, typically, theoretical lower bounds on the estimation performance can be used. With regard to the proposed localization approach of this paper, this section first provides the classic CRLB on the error of the position estimate and second the PCRLB as performance bound for unbiased sequential Bayesian estimators. Thereby, the latter bound allows for evaluating recursive Bayesian filters like the proposed EKF.

### 4.1. Cramér–Rao Lower Bound on Position Estimation

The CRLB is defined as the inverse of the FIM [21]. This means, given the vector parameter $r_{q}$, the elements of the unbiased estimator ${\hat{r}}_{q}={\left[{\hat{x}}_{q},{\hat{y}}_{q}\right]}^{T}$ satisfy
(18)$$\mathrm{Var}\left({\hat{x}}_{q}\right)\geq {\left[F{\left(r_{q}\right)}^{-1}\right]}_{1,1}=\sigma_{x_{q}}^{2}$$
and
(19)$$\mathrm{Var}\left({\hat{y}}_{q}\right)\geq {\left[F{\left(r_{q}\right)}^{-1}\right]}_{2,2}=\sigma_{y_{q}}^{2},$$ where Var(·) denotes the variance of an estimator and the terms ${\left[F{\left(r_{q}\right)}^{-1}\right]}_{n,n}$, n={1,2}, denote the diagonal elements of the inverse FIM $F(r_{q})$.

Inherently, received signals are a function of propagation delays $\tau_{q}$. This holds for every pair of transmitting and receiving nodes in the network. Thus, by applying the chain rule, the FIM $F(r_{q})$ can be alternatively expressed as [21]
(20)$$F\left(r_{q}\right)=J{\left(r_{q}\right)}^{T}F\left(\tau_{q}\right)J\left(r_{q}\right),$$ with the Jacobian matrix $J(r_{q})$ as defined in Equation (Equation 10) and the FIM $F\left(\tau_{q}\right)\in R^{\left|P|\times |P\right|}$ with respect to delay vector $\tau_{q}$ (Please note also that path loss and phase information can be taken into account for calculating the FIM but is out of the scope of this work). For the linear independent time delays $\tau_{q}$, the Fisher information is well known [21] and the diagonal elements of $F(\tau_{q})$ can be written as
(21)$${\left[F\left(\tau_{q}\right)\right]}_{p,p}=\frac{8\pi^{2}\beta^{2}{\mathrm{SNR}}_{pq}}{c^{2}},$$ with $\beta^{2}$ as effective bandwidth of the transmit signal. The index p=1,⋯,|P| provides an enumeration of index set P. Therewith, ${\mathrm{SNR}}_{pq}$ expresses the SNR for the MPC caused by scatterer *q* in network link *p*. With ${\mathrm{Tx}}_{k}$ and ${\mathrm{Rx}}_{l}$ defining link *p*, the SNR can be written as
(22)$${\mathrm{SNR}}_{pq}=\left(P_{{\mathrm{Tx}}_{k}}\frac{G_{{\mathrm{Tx}}_{k}}G_{{\mathrm{Rx}}_{l}}\sigma_{q}c^{2}}{{\left(4\pi \right)}^{3}f_{c}^{2}{\left(d_{kq}^{\mathrm{Tx}}\right)}^{2}{\left(d_{lq}^{\mathrm{Rx}}\right)}^{2}}\right)P_{n}^{-1},$$ where $P_{{\mathrm{Tx}}_{k}}$ and $P_{n}$ denote transmit power and receiver noise power, $G_{{\mathrm{Tx}}_{k}}$ and $G_{{\mathrm{Rx}}_{l}}$ express antenna gains at transmitter and receiver, and $\sigma_{q}$ refers to the RCS of scatterer *q* [7]. Hence, the elements of the FIM in Equation (Equation 21) strongly depend on the position of the scatterer and are proportional to
(23)$${\left[F\left(\tau_{q}\right)\right]}_{p,p}\propto {\left({\left(d_{kq}^{\mathrm{Tx}}\right)}^{2}{\left(d_{lq}^{\mathrm{Rx}}\right)}^{2}\right)}^{-1}.$$

Finally, the CRLB of Equations (Equation 18) and (Equation 19) can be used to evaluate an estimator by comparing it to the RMSE according to inequality
(24)$${\mathrm{RMSE}}_{q}=\sqrt{E\left[∥{\hat{r}}_{q}-r_{q}{∥}^{2}\right]}\geq \sqrt{\sigma_{x_{q}}^{2}+\sigma_{y_{q}}^{2}}.$$

### 4.2. Posterior Cramér–Rao Lower Bound for Nonlinear Sequential Bayesian Estimation

The CRLB, as introduced above, allows for evaluating the positioning performance for a specific time instant. Thereby, the system is assumed to be time-invariant. An evaluation of Bayesian filtering and tracking approaches, however, requires a performance bound, which accounts for time-variant systems, for underlying stochastic state space models, and for the incorporation of prior knowledge [24]. Such a performance bound is provided by the PCRLB. The PCRLB is the theoretical performance bound for sequential Bayesian estimators [24,25]. Equivalently to the CRLB, the PCRLB is calculated by the inverse of the posterior FIM $F_{B,q}\left(t_{n}\right)$ defining the inequality
(25)$$E_{x_{q}\left(t_{n}\right)}\left[\left({\hat{x}}_{q}\left(t_{n}\right)-x_{q}\left(t_{n}\right)\right){\left({\hat{x}}_{q}\left(t_{n}\right)-x_{q}\left(t_{n}\right)\right)}^{T}\right]=M_{q}\left(t_{n}\right)\geq F_{B,q}{\left(t_{n}\right)}^{-1},$$ with $E_{a}\left[\;\cdot \;\right]$ as expectation with respect to the probability density of random variable a. The inequality in Equation (Equation 25) implies that the difference between the MSE matrix $M_{q}\left(t_{n}\right)$ and the inverse of the posterior FIM is a positive semi-definite matrix. Following [24,25,26,27], the posterior FIM $F_{B,q}\left(t_{n}\right)$ can be calculated recursively as
(26)$$F_{B,q}\left(t_{n}\right)=D_{22,q}\left(t_{n}\right)-D_{21,q}\left(t_{n}\right){\left(F_{B,q}\left(t_{n-1}\right)+D_{11,q}\left(t_{n}\right)\right)}^{-1}D_{12,q}\left(t_{n}\right),$$ with
(27)$$\begin{array}{cc} D_{11,q}\left(t_{n}\right)= & E_{x_{q}\left(t_{n-1}\right),x_{q}\left(t_{n}\right)}\left[-\Delta_{x_{q}\left(t_{n-1}\right)}^{x_{q}\left(t_{n-1}\right)}\ln p\left(x_{q}\left(t_{n}\right)|x_{q}\left(t_{n-1}\right)\right)\right], \end{array}$$
(28)$$\begin{array}{cc} D_{12,q}\left(t_{n}\right)= & E_{x_{q}\left(t_{n-1}\right),x_{q}\left(t_{n}\right)}\left[-\Delta_{x_{q}\left(t_{n-1}\right)}^{x_{q}\left(t_{n}\right)}\ln p\left(x_{q}\left(t_{n}\right)|x_{q}\left(t_{n-1}\right)\right)\right]=D_{21,q}{\left(t_{n}\right)}^{T},\\ D_{22,q}\left(t_{n}\right)= & E_{x_{q}\left(t_{n-1}\right),x_{q}\left(t_{n}\right)}\left[-\Delta_{x_{q}\left(t_{n}\right)}^{x_{q}\left(t_{n}\right)}\ln p\left(x_{q}\left(t_{n}\right)|x_{q}\left(t_{n-1}\right)\right)\right] \end{array}$$
(29)$$\begin{array}{cc} \; & +E_{x_{q}\left(t_{n}\right)}\underset{\mathrm{classic}\;\mathrm{FIM}}{\underset{︸}{\left[E_{z_{q}\left(t_{n}\right)|x_{q}\left(t_{n}\right)}\left[-\Delta_{x_{q}\left(t_{n}\right)}^{x_{q}\left(t_{n}\right)}\ln p\left(z_{q}\left(t_{n}\right)|x_{q}\left(t_{n}\right)\right)\right]\right]}}. \end{array}$$

With $\nabla_{a}$ denoting the partial derivatives with respect to a, the operator $\Delta_{b}^{a}=\nabla_{b}\nabla_{a}^{T}$ gives the corresponding second-order partial derivatives. Furthermore, $p\left(a\left(t_{n}\right)|b\left(t_{n}\right)\right)$ is the conditional probability distribution of random variable $a(t_{n})$ given $b(t_{n})$ at time instant $t_{n}$. Note that the term inside the second expectation of $D_{22,q}\left(t_{n}\right)$, as highlighted by a brace, refers to the definition of the classic FIM [25]. For state $x_{q}\left(t_{n}\right)$ given in Equation (Equation 11), the 4×4 FIM is
(30)$$\tilde{F}\left(x_{q}\left(t_{n}\right)\right)=\left[\begin{array}{cc} F\left(r_{q}\left(t_{n}\right)\right) & 0\\ 0 & 0 \end{array}\right],$$ where $F(r_{q}\left(t_{n}\right)$ denotes the FIM of Equation (Equation 20). Thus, referring to the linear transition model introduced in Section 3.3 with transition matrix $A_{q}$ given in Equation (Equation 13) and white Gaussian process noise with covariance matrix $Q_{q}$ given in Equation (Equation 14), Equations (Equation 27)–(29) result in
(31)$$\begin{array}{cc} D_{11,q}\left(t_{n}\right)= & A_{q}^{T}Q_{q}^{-1}A_{q}, \end{array}$$
(32)$$\begin{array}{cc} D_{12,q}\left(t_{n}\right)= & -A_{q}^{T}Q_{q}^{-1}=D_{21,q}{\left(t_{n}\right)}^{T}, \end{array}$$
(33)$$\begin{array}{cc} D_{22,q}\left(t_{n}\right)= & Q_{q}^{-1}+E_{x_{q}\left(t_{n}\right)}\left[\tilde{F}\left(x_{q}\left(t_{n}\right)\right)\right]. \end{array}$$

Inserting Equations (Equation 31)–(33) into Equation (Equation 26) and applying the matrix inversion lemma results in the recursive expression
(34)$$F_{B,q}\left(t_{n}\right)={\left(Q_{q}+A_{q}F_{B,q}{\left(t_{n-1}\right)}^{-1}A_{q}^{T}\right)}^{-1}+E_{x_{q}\left(t_{n}\right)}\left[\tilde{F}\left(x_{q}\left(t_{n}\right)\right)\right].$$

Given prior information according to the probability density function $p(x_{q}\left(t_{0}\right))$, the initial FIM $F_{B,q}\left(t_{0}\right)$ can be calculated as
(35)$$F_{B,q}\left(t_{0}\right)=E_{x_{q}\left(t_{0}\right)}\left[-\Delta_{x_{q}\left(t_{0}\right)}^{x_{q}\left(t_{0}\right)}\ln p\left(x_{q}\left(t_{0}\right)\right)\right],$$ which is used to initialize the recursion in Equation (Equation 34).

## 5. Case Study

In this section, the proposed localization approach is analyzed for the example of a walking pedestrian. Together with the network structure, the measurement setup is presented first. Based on the introduced static network, the localization performance is evaluated theoretically using the performance bounds of Section 4. Finally, the localization approach is applied to channel measurements.

### 5.1. Network and Measurement Setup

The analyzed static network consists of K=1 transmitting and L=4 receiving nodes. As shown in Figure 1a, the network nodes are individually placed with an inter-node distance between 10 m and 15 m. The center of the network forms a transmitting node with a small directional antenna. Thereby, the main beam of the transmit antenna is oriented towards a designated observation area between the receiving nodes, as shown in Figure 1b, illustrating the gain of the transmit antenna in the network area [28]. With the transmit antenna Tx chosen as the origin of the coordinate system, the overall considered observation area spans from −7.5 m to 30 m in the *x*-direction and from −10 m to 26 m in the *x*-direction. The selected scenario considers a walking pedestrian as single mobile scatterer (Q=1), walking a wide circle within the illuminated area of the static network setup. As the experiment was conducted on an apron close to surrounding buildings, an elevated tachymeter was used as ground truth system. Using the tachymeter together with a high precision reflector prism, the individual static antenna locations of the network were determined prior to the experiment. For recording the ground truth during the experiment, the pedestrian wore a helmet, on which the reflector prism was mounted.

As a measurement system, the Medav RUSK-DLR wideband channel sounder was used [29]. Therefore, the measured data are CIR between transmit and receive antennas, i.e., network nodes, respectively. A full summary of the corresponding measurement parameter settings provides Table 1.

### 5.2. Theoretical Performance Evaluation

For illustrating the localization performance of the measurement setup, the RMSE is calculated for the considered observation area according to the CRLB as defined in Equation (Equation 24). To determine the RMSE, the parameters provided in Table 1 are used. Assuming a transmit signal with rectangular power spectral density, the effective bandwidth is defined as $\beta^{2}=B^{2}/12$ [21]. The value of the RCS accounts for an object’s reflectivity characteristics, which is influenced by its size, shape, and material. Thus, for a pedestrian being considered as moving object, a typical RCS is 1 m2 [7]. Eventually, the localization RMSE for the measurement setup is shown in Figure 2a. Overall, the shape of the RMSE reveals good localization performance for scatterers located in areas of high transmit antenna gain; see also Figure 1b. Hence, scatterers located close to the transmitting node can be localized very precisely. Qualitatively, this holds also for scatterers located close to receiving nodes. For these areas, the high localization performance can be explained by the received signal strength of the backscattered signal, since the RMSE strongly depends on the SNR. Additionally, the decreasing localization performance for far positions from the network nodes confirms this observation. Besides SNR, the localization performance depends on the system geometry. Effects due to system geometry can be explained by interpreting the scatterer location as the intersection point of multiple ellipses, which correspond to the propagation delays of respective transmitting and receiving nodes. To achieve high localization performance in both *x*- and *y*-directions, ideally, ellipses should intersect perpendicular to each other. For scatterers located far from the localization network, the shapes of the individual ellipses differ only marginally from each other. Thus, the intersection angle is very low, which results in an increased localization uncertainty in the direction parallel to the tangent at the intersection point. With a widely distributed network covering a large observation area, the effects of low intersection angles can be avoided. A further geometrical effect can be observed for scatterers located in the proximity of the baseline between transmitting and receiving nodes. Regarding a single link, the location information in the direction parallel to the baseline is very low, and, thus, the localization performance also decreases. Accordingly, in the localization network, the localization performance decreases in the proximity of the individual baselines; see Figure 2a. This performance degradation is independent of the network topology.

Please note that the derived CRLB of Section 4 only depends on waveform and SNR. This means that the influence of any superposition of LoS and MPC on the parameter estimation is not considered [27], and is out of the scope of this paper. Since the superposition of LoS and MPCs strongly impacts the estimation capabilities of its influence on parameter estimation will be included in future research.

In order to evaluate the performance of the EKF proposed in Section 3.3, the PCRLB is calculated for different scenarios within the measurement setup and compared to simulation results of the tracking filter. Three different scenarios are considered, each for a single scatterer moving with constant velocity. The initial absolute velocity is per-mode=symbol 1.41
m/s for every scenario. However, each scenario possesses an individual starting position and movement direction. Figure 2b provides an overview over the considered scenarios. Referring to Figure 2a, Scenario I is characterized by high localization capabilities throughout the whole trajectory. Scenario II, in contrast, crosses an area of poor localization capabilities between Tx and ${\mathrm{Rx}}_{3}$. In Scenario III, the trajectory starts in an area of poor localization capabilities and moves towards an area of very high localization capabilities in the main beam the transmitting node. For the calculation of the PCRLB as well as the EKF simulations, similar system and signal parameters as for calculating the static positioning CRLB are used. As process noise intensity for the covariance in Equation (Equation 14), a value of $\sigma_{q}^{2}=$ per-mode=symbol 0.01
m${}^{2}$/s3 is defined [23]. The filter state is initialized randomly according to the initial state covariance $P(t_{0})$ around the initial state $x(t_{0})$ given by the respective scenario. Thereby, $P(t_{0})$ is defined as 4×4 diagonal matrix, with an initial variance of per-mode=symbol 0.1
m${}^{2}$ on position and per-mode=symbol 0.01
m${}^{2}$/s${}^{2}$ on velocity, in both *x*- and *y*-directions. Determining the PCRLB and the EKF performance results requires performing multiple Monte Carlo runs. In each run, the system equations of Section 3.3 are simulated with different samples of the process noise for 10 s, which equals an average walking distance of approximately 14 m. Thus, the multitude of simulation runs allows for approximating the expectation in Equation (33). In order to estimate the MSE matrices in Equation (Equation 25), the measurement noise also needs to be sampled for each run. In this study, for every scenario, 5000 Monte Carlo runs are performed with 50 realizations of measurement noise. Hence, the EKF is evaluated $2.5\times 10^{5}$ times in each scenario. Thereby, the number of Monte Carlo runs was chosen to achieve statistically stable results for the PCRLB and to achieve results for the EKF, which fluctuate only marginally compared to the absolute RMSE values.

Figure 3 shows the simulation results for the three scenarios. The results for both PCRLB and EKF are given in terms of RMSE, i.e., as the square root of the MSE in Equation (Equation 25). The RMSE of PCRLB and EKF are denoted by $\epsilon_{PCRLB}$ and $\epsilon_{EKF}$. Given $P(t_{0})$, the initial RMSE of the positioning error is approximately 0.45
m for all scenarios. Due to sequential filtering characteristics, the localization RMSE strongly decreases shortly after the initialization and follows the static performance capabilities along the respective scenario trajectory. Overall, it can be observed that the filter RMSE values are very close to those of the theoretical bound. This holds for the simulation results of each scenario. Particularly, for Scenario I, the results of the EKF converge very fast and reach the bound in less than 1 s; see Figure 3a. In addition, for Scenario II, the EKF converges to the bound after about 3 s as shown in Figure 3b. With the trajectory crossing an area of poor localization capabilities between Tx and ${\mathrm{Rx}}_{3}$, both the PCRLB and the filter RMSE show a temporary increase. For scatterer positions close to the baseline between transmitting and receiving nodes, the elements of the Jacobian in Equation (Equation 10) and therewith of the observation matrix in Equation (Equation 17) are close to zero. Due to singularities caused by the inversion of the observation matrix, the region close to the baselines between Tx and ${\mathrm{Rx}}_{3}$, as well as Tx and ${\mathrm{Rx}}_{4}$, is particularly challenging for the filter. For calculating the bound in Equation (Equation 34), the expectation of the FIM averages these singularities. The results in Figure 3b confirm this effect, with the filter RMSE diverging from the bound between 1 s and 2 s. Finally, the filter results for Scenario III are provided in Figure 3c. The filter RMSE shows a fast convergence to the bound after approximately 2 s. As described above, however, the effects of singularities on the EKF in regions close to Tx-Rx baselines also influences the filter results of the third scenario. Here, the mean trajectory crosses Tx-${\mathrm{Rx}}_{4}$ and travels along Tx-${\mathrm{Rx}}_{3}$. With increasing state covariance, many simulation trajectories lie on or are close to the baselines. Thus, the filter RMSE does not fully converge to the bound in the period after 2 s of simulation time.

### 5.3. Measurement Based Evaluation

As stated in Section 3, the channel parameters for all links in the network are estimated using KEST. The estimation results of KEST for the measured CIR over time are shown in Figure 4a–d for each link individually. The figures show the consecutive vectors of delay estimates in Equation (Equation 5) over the full measurement time. Static delay estimates, including LoS and static MPCs, are shown in gray. These static delays together with corresponding standard deviations are determined during a preceding calibration phase according to the procedures described in Section 3.1. This characterization of the static propagation environment allows for extracting the mobile MPC, highlighted in color according to the estimated amplitude level. For each link, the delay estimates fit the ground truth data as indicated by black dashed lines. Hence, the results qualitatively confirm the point scattering assumption for pedestrians [29]. Due to the geometrical arrangement of the receiving antennas and limits in the dynamic range of the measurement system, the detection, estimation, and tracking capabilities of KEST differ. With the small directional antenna gain and the orientation of the transmitting antenna, as shown in Figure 1b, the LoS signal power is reduced for all links. The reduced LoS signal power avoids an elevation of the noise floor due to limits of the dynamic range. Noticeably, link Tx-${\mathrm{Rx}}_{3}$ exhibits particularly good parameter estimation throughout the measurement. On the one hand, this can be explained by the advantageous placement of the receive antenna with respect to the transmit antenna gain, i.e., very low Tx gain towards ${\mathrm{Rx}}_{3}$. On the other hand, ${\mathrm{Rx}}_{3}$ and also ${\mathrm{Rx}}_{4}$ are not impacted by strong multipath fading like ${\mathrm{Rx}}_{1}$ and ${\mathrm{Rx}}_{2}$. Since ${\mathrm{Rx}}_{1}$ and ${\mathrm{Rx}}_{2}$ are located closer to the hangar wall, signal reflections off the wall are received with higher power. Examples for such multipath fading provide the estimation results of ${\mathrm{Rx}}_{1}$ and ${\mathrm{Rx}}_{2}$, where double reflections from the pedestrian and the hangar wall are clearly visible (see Figure 4a,b). However, apart from the perturbing effect regarding parameter estimation, assignable double reflections from mobile scatterers would contain additional location information. Even though in this work double reflections are not considered as source of information, an exploitation of reflected mobile MPC will be included in future research. Besides dynamic range and multipath fading, the estimation results of Figure 4 clearly show how the presence of LoS and static MPCs impact the composition of measurement vector ${\hat{\tau }}_{q}$ and therewith the localization of the mobile scatterer. An unambiguous solution of Equation (Equation 8) requires at least three independent measurements. The required static environment mitigation, i.e., the displacement of any delay estimates close to static components (see Section 3.2), however, reduces the amount of delay estimates assigned to a specific mobile scatterer. Thus, a rich static MPC environment reduces the overall localization capabilities with the current approach. This holds particularly for very sparse networks, such as the four-link network considered in this paper, since an outage in two links impedes localization. Besides static MPCs, the LoS also impacts the localization capability due to the so-called blind zone problem [30]. This means that MPC caused by scatterers located close to the baseline between a transmitting and receiving node is hardly detectable. Exemplary, the parameter estimation results for link Tx-${\mathrm{Rx}}_{4}$ confirm these blind zones, since it is not possible to extract mobile MPCs when the pedestrian is located in the proximity to the link baseline. Thus, for ${\mathrm{Rx}}_{4}$, no parameter estimates are available from 33 s to 44 s.

Finally, the localization approach proposed in Section 3 is used to estimate and track the location of the walking pedestrian using the CIR measurements described above. For initialization, the iterative localization procedure of Equation (Equation 9) is applied to determine an initial location estimate based on available parameter estimates of the four links at $t_{0}=0$ s. The initial velocity components in *x*- and *y*-directions are randomly chosen and taken from a uniform distribution U(0 m/s,1 m/s), respectively. Given the initial state, the EKF described in Section 3.3 is used for tracking the location of the pedestrian. According to [23], the covariance matrix of the process noise in Equation (Equation 14) is determined by the process noise intensity of $\sigma_{q}^{2}=0.01\;m^{2}/s^{3}$. The resulting positioning error for the experiment over time is given in Figure 5 and the corresponding CDF in Figure 6. Overall, except for a few outliers, the positioning error remains below 1 m throughout the experiment. In particular, according to the CDF, more than 97% of positioning errors are below 1 m. The first outlier appears between 20 s and 25 s. This time period coincides with a sharp left curve, as shown in Figure 1a. Thus, a delayed adaption of the velocity vector by the EKF can be a possible explanation for the exceeding positioning error. A very strong localization performance can be observed between 32 s and 37 s, when the pedestrian passes the main beam of the transmit antenna. Due to strong reflection and therewith high SNR, the parameters of each link and thus the position can be estimated very well. For the period from 37 s to 44 s, the error steadily increases. In this period, parameter estimates are almost solely available for ${\mathrm{Rx}}_{3}$. Hence, the few simultaneous and even erroneous delay estimates can explain the increasing positioning error and the outlier at about 44 s. After leaving the blind zone of Tx-${\mathrm{Rx}}_{4}$ at about 44 s, the additional link parameter estimates support the positioning performance thereafter. The increasing error after 53 s is again due to few simultaneous parameter estimates from different network links. Similarly to before, the position estimation is mainly driven by parameter estimates for ${\mathrm{Rx}}_{3}$.

## 6. Conclusions

This paper presented a localization approach to detect, locate, and track moving, non-cooperative objects by means of a network of spatially distributed transmitting and receiving nodes. With moving objects affecting the radio spectrum as time-variant MPC, i.e., discrete mobile scatterers, multipath propagation could be exploited for passive localization. Therefore, the localization of mobile scatterers was formulated as a nonlinear optimization problem. For initialization, the proposed approach uses an iterative nonlinear least squares algorithm following Levenberg and Marquardt to solve the optimization problem. Subsequently, an EKF is applied to estimate the scatterer location recursively over time. For both the snapshot based localization and the nonlinear sequential Bayesian estimation problem performance bounds, the classic CRLB on position estimation and the PCRLB were derived.

The proposed approach is evaluated based on a case study, which considers a pedestrian as a single mobile scatterer walking within a network of one transmitting and four receiving nodes. Thereby, a simulation study has shown that the EKF achieves a localization performance very close to the PCRLB. Even though the EKF performance results diverged from the theoretical bound in areas close to Tx-Rx baselines due to singularities, the study overall confirmed the applicability of the EKF for the localization problem. Particularly, for scatterers remote from any Tx-Rx baseline, the localization performance of the filter was shown to converge to the PCRLB. Moreover, the approach was applied to wideband measurement data corresponding to the case study of the theoretical analysis. With a resulting localization accuracy of 0.8 m at 90% confidence, the case study proved that mobile scatterers can be localized using CIR, i.e., exploiting multipath propagation.

The performance of the proposed approach strongly depends on the underlying parameter estimation algorithm and the mitigation of the LoS and static MPC. Since both the parameter estimation and the mitigation of the static components are prone to errors, future work will focus on a direct multilateration approach. The direct usage of CIR for localization and tracking could help to overcome the dependence on individual parameter estimation and would link an object’s mobility information implicitly to the recursive position estimation.

## Figures and Tables

**Figure 1 sensors-19-04802-f001:**
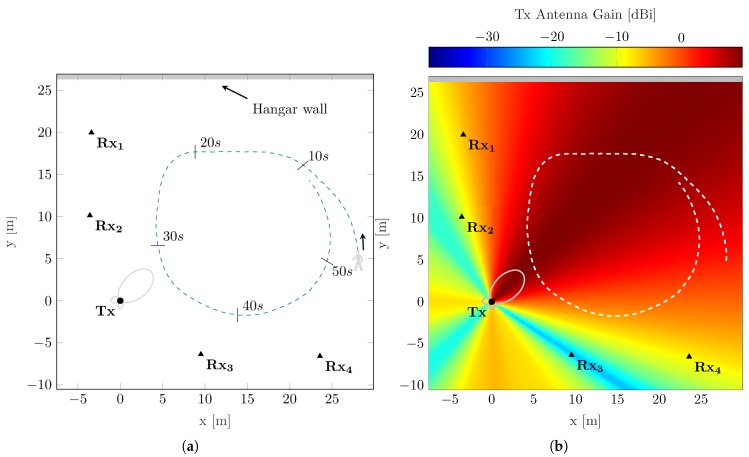
Overview of the evaluated measurement network with transmitting and receiving nodes as circle and triangles. The dashed lines illustrate the trajectory traveled. The gray line highlights the hangar wall as the strongest static reflector in the environment. The vertical radiation pattern of Tx antenna is shown in light gray. (**a**) Scenario overview with starting position of moving pedestrian and movement direction; (**b**) Scenario overview illustrating gain of small directional transmit antenna [28].

**Figure 2 sensors-19-04802-f002:**
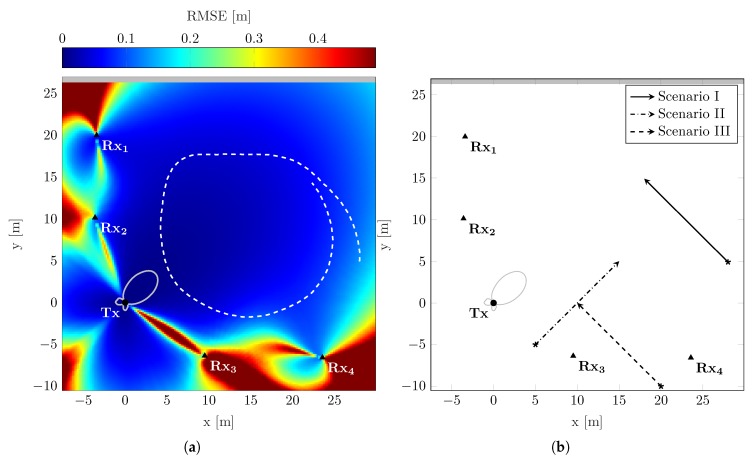
Overview of the measurement network, showing (**a**) the resulting CRLB on position estimation with dashed line as experiment trajectory, and (**b**) the three scenario trajectories for evaluating the PCRLB and the EKF; the trajectories represent the mean values of the position over a time period of 10s and indicate starting positions and moving directions.

**Figure 3 sensors-19-04802-f003:**
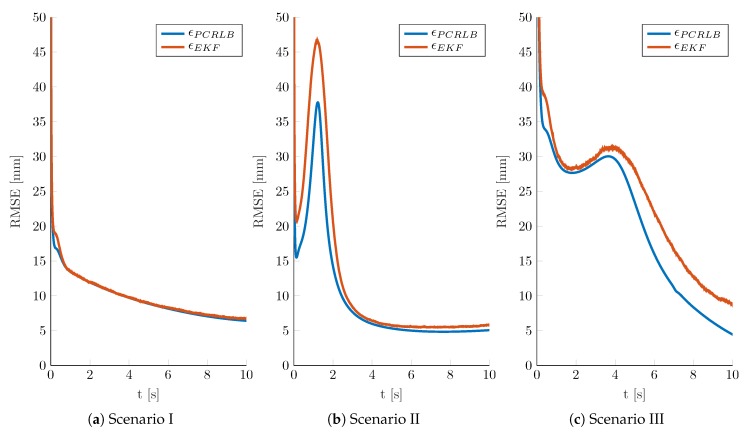
Simulation results for Scenarios I–III as provided in Figure 2b. Results for both PCRLB and EKF are given in terms of RMSE, referred to as $\epsilon_{PCRLB}$ and $\epsilon_{EKF}$.

**Figure 4 sensors-19-04802-f004:**
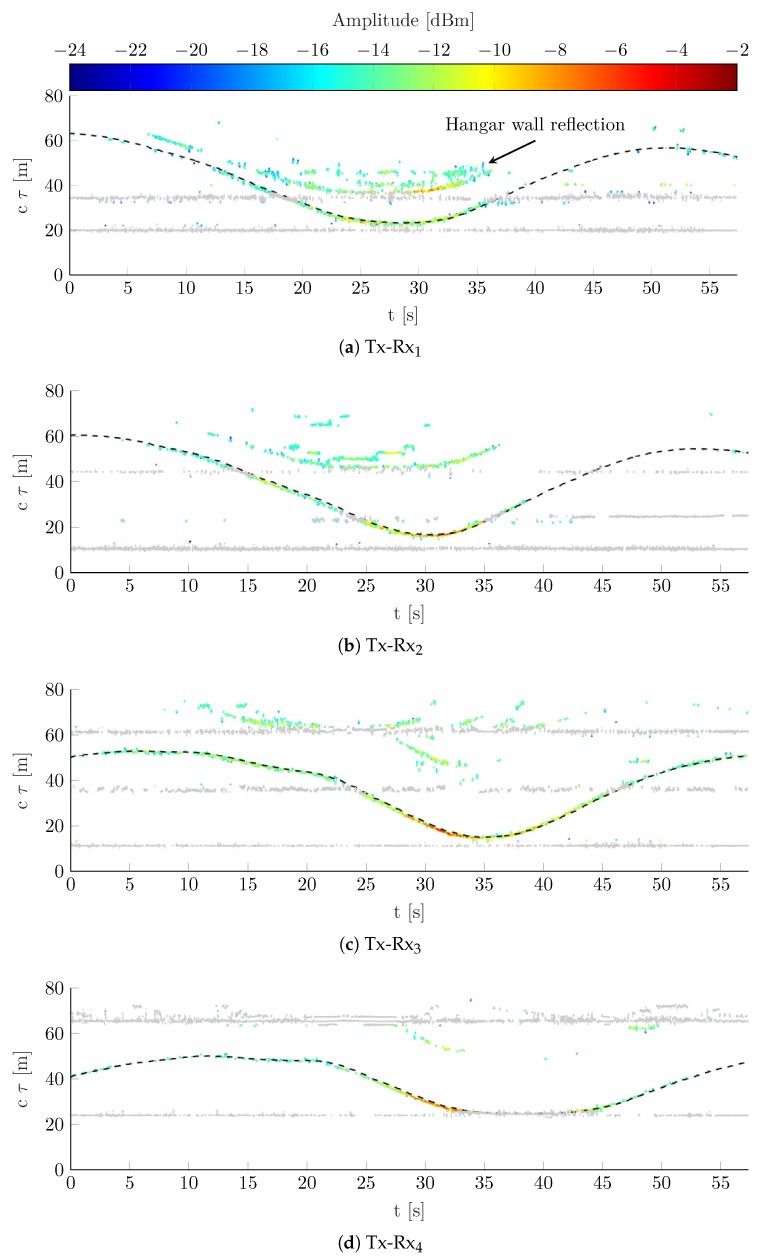
Estimation results of KEST for CIR over time of moving pedestrian. Extracted mobile MPC is colored according to the estimated amplitude level. LoS and static MPC are shown in gray. Dashed black lines indicate delays from ground truth data. Other mobile MPC deviating from ground truth can be referred to double reflections of moving pedestrian and hangar wall.

**Figure 5 sensors-19-04802-f005:**
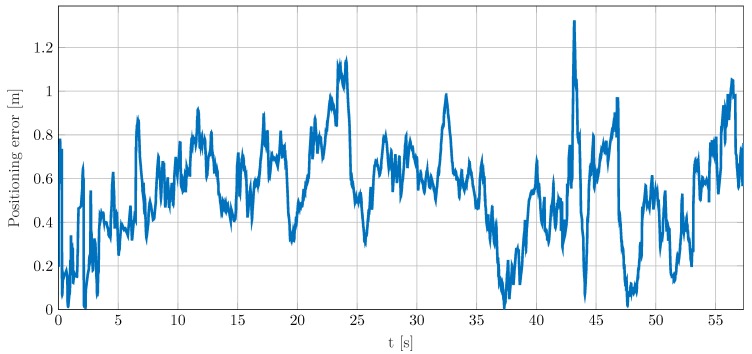
Positioning results of localization approach based on channel measurements—absolute positioning error of moving pedestrian over time.

**Figure 6 sensors-19-04802-f006:**
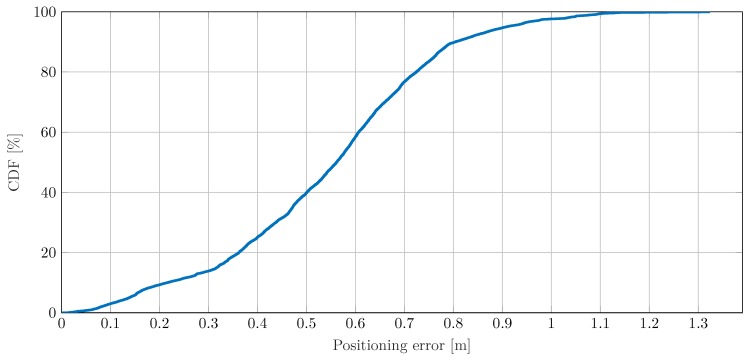
CDF of absolute positioning error for a moving pedestrian scenario.

**Table 1 sensors-19-04802-t001:** Measurement parameters.

Parameter	Value
Center frequency $f_{c}$	5.2 GHz
Bandwidth *B*	120 MHz
Signal period $T_{p}$	3.2 μs
Measurement rate $T_{g}$	2.048 ms
Transmit power $P_{\mathrm{Tx}}$	37 dBm
Antenna gain $G_{\mathrm{Tx}}$	9 dBi (small directional [28])
Antenna gain $G_{{\mathrm{Rx}}_{1-4}}$	8 dBi (toroidal, omni-directional)

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
