# Peer review of "Localization and Tracking of Discrete Mobile Scatterers in Vehicular Environments Using Delay Estimatesâ€"

_sensors, 2019, doi:10.3390/s19214802_

Round 1
Reviewer 1 Report
From the reviewer point of view, the paper is well written, the objectives are very clear, the bibliography review is broad, and the applied methodology and the results seem to be interesting. Especially interesting are the results in section 5, where experiments in real scenarios are done. There, authors show some of the real practical problem in localization and tracking evaluation, as signal reflections.
In order to improve this good paper, some more practical consideration could be added who could help further interested readers. For example, authors decomposed the procedure in three stages: calibration, estimation and tracking; then, what happens if conditions change after calibration, for example, studying the influence of some metal structure included in the scenario, who changes considerably the electromagnetic field).
Reviewer 2 Report
This paper proposes a very interesting and focused work on the localization (and tracking) of mobile agents in a confined environment.
The paper looks well written and organized. Therefore, only minor improvements are suggested according to the following:
(2): though rather obvious, it would be nice to also include right after a description of the delta appearing in the formula. (12) describes a very simple dynamics for the scatterer. Accounting for standard descriptions of particle dynamics, even away from random walk, it would be better to frame this assumption/description. Line 276: check grammar. Lines 281-287: concerning the possible “poor system geometry” issues left for “future research”, it would be nice to read here something related to possible suggestions on how to improve/optimize the network topology to avoid or reduce to a minimum such issue. Lines 299, 376: the measurement units for the variance sigma_q^2 may look rather unusual, so it would be good to add a comment on it. Line 307: as for the mentioned number of runs of the Monte Carlo analysis, it is requested to add some results/data to prove the convergence of the sought statistics after 5000 runs.Author Response
Please see the attachment.
